# “Like Death is Near”: Expressions of Suicidal and Homicidal Ideation in the Blog Posts of Family Caregivers of People with Dementia

**DOI:** 10.3390/bs9030022

**Published:** 2019-03-03

**Authors:** Joel G. Anderson, Alexis Eppes, Siobhan T. O’Dwyer

**Affiliations:** 1College of Nursing, University of Tennessee, Knoxville, TN 37996, USA; lex.eppes@gmail.com; 2Medical School, College of Medicine and Health, University of Exeter, Exeter EX1 1TX, UK; s.odwyer@exeter.ac.uk

**Keywords:** family caregiver, suicide, homicide, dementia, Alzheimer’s disease, blog

## Abstract

Background: The challenges of providing care for someone with Alzheimer’s disease and related dementias (ADRD) have been associated with increased stress, poor mental and physical health, social isolation, and financial distress. More recently, caregiving has been associated with high rates of suicidal and homicidal ideation, but the research on these phenomena is limited. The present study analyzed a sample of blogs written by family caregivers of people with ADRD to explore thoughts of suicide and homicide expressed by these caregivers. Methods: Blogs written by self-identified informal caregivers of people with ADRD were identified using a systematic search method and data were analyzed using a qualitative thematic analysis. Results: Five themes related to thoughts of suicide and homicide by caregivers and people with ADRD were derived from the analysis: (1) end-of-life care; (2) thoughts of death and euthanasia by the person with ADRD; (3) surrogate decision making; (4) thoughts of suicide by the caregiver; and (5) thoughts of homicide and euthanasia by the caregiver. Conclusions: The results capture the reality of suicidal and homicidal thoughts among family caregivers of people with ADRD, supporting calls for more research on these complex topics and highlighting the need for changes to clinical practice to prevent thoughts from becoming behaviors or actions.

## 1. Introduction

Unpaid family caregivers are the largest source of long-term care for the ever-increasing number of people with Alzheimer’s disease and related dementias (ADRD) [1]. These caregivers must cope with a progressive decline in the person’s cognitive capacity, communication skills, and functional abilities, and learn new skills to manage the behavioral and psychological symptoms that are common in dementia [2,3]. These challenges contribute to a range of negative physical and mental health outcomes [4,5], as well as notable increases in caregiver morbidity and mortality [6].

Recent literature also suggests that family caregivers might be at high risk of suicide, though the research to date is limited. Suicidal ideation experienced by family caregivers has been examined among men caring for male partners with HIV [7], as well as people providing care for family members with chronic disease or disability [8,9]. However, the combination of the physical and cognitive decline is likely to make caring for someone with ADRD more burdensome than other disabilities/illnesses [10]; therefore, a particular focus on this population of caregivers is warranted. Two recent studies report that between 12% and 16% of dementia caregivers have experienced suicidal thoughts [5,11,12]. Although these studies suggest that caregivers are contemplating suicide at up to four times the rate of the general population, the purely quantitative approach fails to capture the nuance and lived experience of suicidal thoughts and behaviors among family caregivers.

In addition to suicidality among caregivers, there is also a small body of evidence on homicidal thoughts and behaviors in this population. Studies of media reports and mortality databases have identified homicides and homicide–suicides perpetrated by people caring for family members with a range of illnesses and disabilities, including ADRD (e.g., [13,14,15,16,17]). Although these studies provide strong evidence of a clear, persistent, and global phenomenon, they are necessarily retrospective and can only speculate about potential motives and catalysts. Research that examines homicidal thoughts in living caregivers may yield greater insights, but there has been only one previous qualitative study of homicidal thoughts in family caregivers of people with ADRD [18].

As the population ages and rates of dementia rise, the need for family care will grow exponentially. A better understanding of suicidal and homicidal thoughts and behaviors in family caregivers, including how they develop and change over time [5], will be essential for ensuring at-risk caregivers can be identified and supported. Unfortunately, conducting primary research on suicidal and homicidal thoughts and behaviors is both ethically and practically challenging given the sensitive nature of the topic, a lack of understanding amongst institutional review boards and other gatekeepers, limited measurement tools, variations in nomenclature, and differences in reporting requirements and data management legislation across locations [19,20]. Existing sources of data, however, may be able to provide powerful insights. Social media, for example, has already been shown to offer valuable evidence of a range of health behaviors and outcomes [21], including for family caregivers [22,23]. 

The term social media refers to a set of Internet-based platforms that allow users to create content [24]. Among these social media platforms, web logs (now more commonly referred to as blogs) are frequently used as online journals to document the life experiences of individuals and groups. Within the blog genre, illness blogs provide rich narratives of the lived experience of various chronic and terminal conditions [25,26]. Similarly, blogs written by family caregivers offer detailed accounts of the experience of providing care for people with ADRD. As well as being a valuable source of catharsis, community, and support for those who blog and those who read blogs [27,28,29,30], caregiver blogs also provide a rich source of data for researchers [22,23]. To our knowledge, however, there has been no previous research on suicidal and homicidal ideation in the blogs of family caregivers of people with ADRD. The aim of this study was to examine expressions of suicidality and homicidality in the blogs of people caring for family members with ADRD. 

## 2. Materials and Methods

### 2.1. Study Sample and Data Collection

To locate blogs for analysis, a purposive sampling approach was taken. A Google search was conducted, using the search terms “dementia”, “caregiver”, and “blog”. Blogs were included in the study sample if (1) the blog author described themselves as a family caregiver of a person with ADRD, (2) the blog was written in English, and (3) the majority of blog posts were related specifically to the experience of providing care for a family member with ADRD. For this last inclusion criterion, the first and last posts from each month of the most recent two years of the blogs were read to determine if the author was writing primarily about their experience as a caregiver. Once selected for inclusion in the study sample, all posts related to the caregiving experience were extracted for analysis. Blogs maintained by associations, organizations, or health care service providers were excluded from the sample given that these blogs are not meant to serve solely as first-person accounts of caregiving. Additionally, blogs that consisted merely of links to resources or that contained no description of individual caregiving experiences were excluded. From the initial search, ten blogs were identified for the analysis. One blog written by a caregiver living in India was excluded from the sample because the remaining blogs were written by individuals living in Western countries (the United States and the United Kingdom) whose experiences in caregiving are likely to be similar given the social, cultural, and political context of health and care services in these countries.

Approval to conduct the study was granted by the Institutional Review Board. The research team was mindful of the perceived privacy of the bloggers. Internet users generally understand the risks of publishing narratives online and recognize that their narratives may be repurposed in ways different than the original purpose [31]. Because the blogs included in the study sample did not require group membership to access and were not password-protected, informed consent was not considered necessary or sought [32]. Bloggers were, however, notified that their blogs were being included in a research study and they were given the opportunity to opt out. If an email address was provided on the blog, an email outlining the purpose of the study was sent to the blogger. If no email address was provided, the same information was posted as a comment to the most recent post on the blog. None of the bloggers asked to have their blog omitted from the analysis. Of the nine bloggers, only two bloggers responded to the opt-out emails/comments. Both responded favorably to the inclusion of their blogs in the analysis. 

### 2.2. Data Analysis

From the nine included blogs, there were 2345 blog posts eligible for analysis. These were exported into Microsoft Word to create transcripts. The transcripts were imported into NVivo software and analyzed individually and in aggregate to understand the individual perspective of each blog and the sample as a whole. To protect confidentiality, all personal identifiers within the transcribed blog posts were removed before analysis. Demographic data were gleaned from the text of posts (e.g., a caregiver describing their relationship to the care recipient). A thematic analysis was conducted [33] as in previous analyses [22,23]. Transcripts were analyzed in chronological order by two authors (JGA and AE) using codes created from the verbatim words used by the bloggers. Definitions for codes were agreed upon through consensus among members of the research team. Analytic memos, including thoughts from the researchers in terms of keywords and sentences, were maintained throughout the data analysis to aid in identifying categories and themes evident in the data. 

The research team met several times to compare results during the process of data analysis, iteratively assessing the codes and categories that emerged. No further analysis was attempted once no new codes were being created; saturation was reached with regard to the primary research question. Groups of codes that expressed similar ideas or phenomena were grouped into categories. Similar categories were then merged to refine interrelationships and to establish major themes. Findings were discussed among members of the research team to reach consensus regarding these themes. Trustworthiness of the data analysis was addressed by having all components of the study design and data analysis open to review, as well as using the reflective notes and the iterative process of the data analysis [34]. 

Finally, although the caregivers blogged about a range of issues and topics, for the purposes of this paper we focused only on expressions of suicidal and homicidal ideation (other issues and topics have been reported elsewhere [22,23]). We defined these concepts broadly to include passive and/or active thoughts of death for the caregiver or the care recipient.

## 3. Results

Data were extracted from a sample of nine blogs written between 2012 and 2015 by self-identified caregivers of people with ADRD. The majority (*n* = 7) of caregivers were the daughter of the person with ADRD, with the remainder being heterosexual spouses/partners. The majority of caregivers (*n* = 8) and care recipients (*n* = 6) were women. Five themes were identified in the data: (1) end-of-life care; (2) thoughts of death and euthanasia by the person with ADRD; (3) surrogate decision making; (4) thoughts of suicide by the caregiver; and (5) thoughts of homicide and euthanasia by the caregiver. In addition to the frequency of the individual codes used across the dataset, the number of blogs referenced for each code from the full sample is provided in Table 1. No suicidal or homicidal behaviors or acts were described in the blog narratives.

### 3.1. End-of-Life Care

When writing about issues and thoughts related to death, caregivers frequently seemed to be grappling with the prospect of the death of the person with ADRD and the challenges of providing end-of-life care. The majority of caregivers wrote about what they described as “the long good bye”. Frequently, caregivers expressed unsettling feelings and thoughts related to the decline and inevitable passing of their family members, wondering if they could continue to provide care and whether the person’s quality of life could be sustained. One caregiver wrote, “It is now at the point where I look at her and find myself wondering how long. That in itself is such a huge question with so many feelings”.

In contrast to the long goodbye, some caregivers described feeling that the end of life was catching them off guard. Numerous posts reported feeling “like death is near” and expressed surprise that events occurred more rapidly than anticipated. One caregiver wrote, “I am still in a bit of shock as I did not think we would get here this quickly”. The speed with which the person with ADRD deteriorated was a particular cause of caregiver angst related to end-of-life care. Several caregivers expressed that while some conversations and plans regarding the end of life may have been in place, the perceived sudden need to make decisions left caregivers doubting the right course of action.

As caregivers wrote about these end-of-life issues, the concept of dignity and the question of what constitutes a “dignified death” often arose. Caregivers expressed the desire for an idyllic passing for their loved ones, one “free of pain and suffering”. Despite this, many shared anecdotes that led them to believe the exact opposite would happen.
It seemed much easier to think she may die with dignity rather than this terrified, neurotic, miserable person that is unable to do essentially anything.

### 3.2. Thoughts of Death and Euthanasia by the Person with ADRD

In the vast majority of blog posts related to death and dying, caregivers documented the person with ADRD’s thoughts and reflections on their own impending death. One caregiver wrote, “She told me she is dying and everything is hard”. Another caregiver said, “Another new norm is Hubby’s declaration of his death. He talks about it quite often”. Frequently, caregivers described these discussions as arising after the “meds wear off” or at times when the symptoms of dementia were most severe.
She decided to read the Bible with me […] She attempts to read a word or two and starts screaming and crying. Completely distraught over her disease and wishing she would die. My heart breaks for her to see her like this.

Often, caregivers wrote of their care recipients’ explicit desires for death. According to the blogs, these desires were expressed more frequently as the disease progressed and the person with ADRD neared the end of life.
At the end of my father’s life, his misery was so profound, all he wished for was death. He told me that every time he woke up, he was disappointed.
“Let’s get on with this”, she says, matter of factly *[sic]*. “Take me to the mausoleum”.

These declarations of death and dying by the people with ADRD had a negative effect on the caregivers, as well as other members of the family, particularly the children of the caregiver. For example, while caregivers wrote of the positive relationships a grandparent with ADRD might have with grandchildren, these relationships became more complicated when the desire for or anticipation of death was shared with young grandchildren.
Everything was going well […] except when my mother decided to tell [my young daughter] that her brain is dying and she will be dead in a year.

Caregivers also described situations in which the person with ADRD acted on their desire to die. These situations often involved the person with ADRD refusing to eat or take their medications, as well as refusing medical care for emergent issues or other chronic conditions. One caregiver wrote that her husband with ADRD was “[…] not taking his meds in hopes the process might speed up”. Some caregivers also reported implicit and explicit requests to help the person with ADRD die.
She is still looking at me intently, as if I’m hiding the key that will grant her efficient passage out of this world.
Tonight as I drove home, I was thinking about her begging me for the right to die.

### 3.3. Surrogate Decision Making

Caregivers often wrote about concerns surrounding surrogate decision making. In their capacity as health care surrogates either through formal (e.g., power of attorney) or informal arrangements, caregivers expressed the angst that accompanied these decision-making processes. This anxiety was often related to a sense that there was no clear answer, “no right answer”, when making these decisions.
I could change my mind any time, but sending her to the hospital would mean aggressive, life-extending treatment […] This was not black or white. I hit the gray line. I was prepared to let Mom die if it was clearly her time. But here I was, unsure.

Several caregivers described discussions of these surrogate decision-making processes, either in the moment or in the past. These conversations appeared to occur frequently throughout the caregiving process, with caregivers describing conversations with other family members, most often siblings, and health care providers about what to do next. This was especially true in emergent situations, in which a serious infection or fall resulted in the person with ADRD being taken into emergency care or admitted to hospital. Caregivers often wrote about knowledge of their family members’ wishes surrounding end of life, which informed their surrogate decision-making process. Sometimes, this happened in the moment, with one caregiver describing the moment that her mother said, “‘I don’t want any life support whatsoever. You just let me die’. She says that now [in the emergency room]”. Despite this knowledge, the process was not made easier by medical professionals who seemed to prioritize length of life over quality of life.
But I know my mom, and she discussed these issues when we were kids. She never wanted to be a burden, never wanted any lifesaving measures, and she wanted to go when it was her time. So how do I let Mom go when the doctors can always do more?

Caregivers found themselves in these situations often, which seemed to magnify the impact of the process. “Telling the nursing home not to do anything extraordinary to keep her alive is a challenge. ‘Comfort measures only’, is my repeated request”. Additionally, the active thoughts related to death and actions taken by the person with ADRD complicated the process of surrogate decision making, despite the care recipient’s wishes.
Fortunately, Mom was very clear about her DNR order in writing on her will, and also specified no tube feeding, etc., but such documents don’t address how we can act when our loved one feels ready to die intentionally, by refusing meds or further treatment, in a state that could be considered “out of one’s right mind”.

### 3.4. Thoughts of Suicide by the Caregiver

Caregivers expressed thoughts related to their own death. Frequently they reframed these thoughts to avoid explicit suggestions of suicide, such as in the following quote: “I didn’t have thoughts of suicide, but I did think about how much better it would be if I just dropped dead”. Caregivers wrote about how their own death would be an end to the caregiving experience and its struggles. One caregiver stated, “Drifting off blissfully to oblivion in my sleep, unassisted, is a gift I dare not even consider”. Most often these expressions were a declaration by caregivers that they did not want to go through the same experiences as their care recipients. For example, one caregiver said, “Of course. I know exactly what I’ll do. I won’t hang around. When I stop being the person I value being, I will end it”. Another stated the following: “And if I ever get a second opinion indicating the imminent demise of my brain, I will go before it’s too late”.

### 3.5. Thoughts of Homicide and Euthanasia by the Caregiver

Caregivers in this sample of bloggers also wished for the care recipient’s death. They frequently justified these thoughts by noting that the care recipient was “soon going to die anyway”. The vast majority of these thoughts were passive—such as this quote from a caregiver who wrote “I cried on the drive home and selfishly wished for my mom’s end to come soon”—and all sought an end to active caregiving, particularly as the care recipient declined cognitively or physically and the demands and strain of caregiving increased. However, some expressed clear plans for the care recipient’s death, with one caregiver saying she wanted to let her mother die “[…] to let her die—to let her stop all her meds and ‘get this over with’”. Other times, caregivers made direct references to euthanasia. One caregiver stated the following: “If she was a pet, I’d have taken her to the vet and ended it by now. And it would be the right thing. Unfortunately, it’s not the LEGAL [*sic*] thing”. 

## 4. Discussion

The purpose of this study was to understand the experiences of family caregivers and their thoughts related to suicide and homicide, as expressed in blogs. Through thematic analysis of the nine blogs in this study sample, five themes were derived: (1) end-of-life care; (2) thoughts of death and euthanasia by the person with ADRD; (3) surrogate decision making; (4) thoughts of suicide by the caregiver; and (5) thoughts of homicide and euthanasia by the caregiver. These themes are consistent with previous research on suicidal and homicidal ideation among family caregivers [5,11,12,18]. More importantly, however, they also extend our understanding of these phenomena by exploring how caregivers share their experiences in public fora. Although in this analysis we were looking for content related to suicide and homicide, the blog posts were not limited to those topics, with other areas of the caregiving experience having been explored previously [22,23] and in ongoing analyses. These blog posts exquisitely illustrate how complex caregiving issues are and how these intersect with broader issues of death and dying. It is particularly important to highlight the fact that suicidal and homicidal thoughts expressed in the blog posts were not explicit (perhaps out of fear), but still were quite open, which is a testament to the importance of exploring and discussing these issues.

In previous research, roughly 13% of family caregivers of people with ADRD reported suicidal thoughts as part of a two-year, longitudinal study in the Netherlands, with a third reporting suicidality at multiple time points over the two-year time period [5]. Similarly, a cross-sectional Australian study found approximately 17% of family caregivers of people with ADRD reported suicidal thoughts [12]. Previous research may, however, have underestimated the rate of suicidal ideation in this population. Joling and colleagues [5], for example, only asked about suicide in caregivers who screened positive for symptoms of depression, but there is clear evidence in other studies that not all suicidal caregivers are depressed [11]. In our study, all caregivers expressed thoughts related to suicide through expressions of their own death or suicide, as well as thoughts of “giving up” on life. This is particularly compelling, given that these caregivers were sharing their experiences in a public forum, not in the confidential context of a typical research study. It is also consistent with the idea of “hidden ideators”, or those who are at risk of attempting suicide but do not disclose their thoughts or plans in formal settings [35], which underscores the importance (in both research and clinical practice) of asking caregivers directly about thoughts of suicide.

For the purposes of the current study, thoughts of suicide, self-harm, or death were grouped together under the concept of suicidality [36]. There remains debate as to whether these exist on a spectrum of suicidal phenomena [8] or whether self-harm and general thoughts of death are distinct experiences unrelated to suicide [37]. Regardless of theoretical distinctions, and the absence of suicidal acts in the current study, it is essential that any thoughts of suicide, self-harm, or death are taken seriously, as these thoughts reflect a level of distress that is not being addressed [38]. 

Nearly half the caregivers in the present study also expressed thoughts of wishing the death of their care recipient to be hastened, though no evidence of homicidal behaviors was described. This finding is consistent with previous research that identified active thoughts of homicide, passive thoughts of death, and thoughts of euthanasia among family caregivers [18]. However, the prevalence of such thoughts in the current study is much higher than previous research (compelling again because of the public nature of these data). Our results emphasize the reality of homicidal thoughts among family caregivers, supporting the need for additional research and potential changes to clinical practice. In an effort to prevent thoughts from becoming behaviors or actions, identification of caregivers experiencing these thoughts, as well as the development support services for these individuals, is imperative [12,39], particularly in settings where the end of life is near and high-quality palliative care is not being provided for the person with ADRD. Given the relative infancy of this field it is difficult to make more specific recommendations for actions that can be taken in practice and policy. Future research exploring the support preferences of suicidal and homicidal caregivers could make a significant contribution here.

Additionally, the findings underscore the complexity of the issue in terms of nomenclature and defining what is meant by homicide and euthanasia [18]. The caregiver bloggers in the current dataset all resided in countries without legalized euthanasia (eight in the United States, one in the United Kingdom); euthanasia is currently legal in five countries (Belgium, Canada, Colombia, Luxembourg, and the Netherlands [40]). The so-called “right to die” drives much of the debate regarding assisted dying and is related to democratic principles of autonomy [41]. Other cultural aspects must also be taken into consideration, particularly the availability of firearms in the United States. Previous research concerning homicidal ideation expressed by caregivers of people with ADRD has been conducted in countries with strict gun control legislation [5,18]. Societal shifts favoring a more liberal culture may increase approval of euthanasia [42]. For example, in the Netherlands, one-third of physicians, 58% of nurses, and 77% of the general public agree with euthanasia in cases of severe dementia [43]. Caregivers surveyed in the Netherlands supported access to assisted death in people with ADRD, wanted the option of assisted death for themselves were they to be diagnosed with ADRD, but felt unable to make the decision on behalf of another [44]. Individuals may fear unmanageable pain and psychological suffering at end of life [44], thus seeing assisted death as a measure to prevent suffering [45]. These cultural factors may all play a role in how caregivers process their thoughts related to suicide and the death of their care recipients, as well as their approaches to end-of-life care. More research is needed to understand how these cultural aspects have an impact on and approach to the caregiving process, from both the standpoint of the caregiver and the person with ADRD. Comparative research across countries may also help to identify how various policy initiatives (including gun control, legalized euthanasia, and caregiver support services) are contributing to experiences of suicidal and homicidal ideation and inform recommendations for improved policy to better support family caregivers.

Appropriate hospice and palliative care can help to alleviate stress at end of life, for both patients and families, but the quality of and access to end-of-life care for persons with ADRD is often poor and inequitable [44]. In the current study, it was clear that caregivers were grappling with the challenges of providing quality end-of-life care and that this may have contributed to thoughts of suicide, homicide, and euthanasia. Caregivers may experience feelings of distress, depression, anxiety, guilt, and anticipatory grief while providing care for someone with ADRD [46,47,48], particularly when faced with a sense of being unable to cope with the demands of caregiving or during end-of-life decision making [49,50,51]. Limited research to date has explored the experiences of caregivers of people with ADRD with regard to death, providing limited insight into aspects of the impact of disease progression, end-of-life decision making and care, and the impact of social support [52,53,54].

Caregiver comments in the current study also highlight the complexity of end-of-life decision making and advanced directives for people with ADRD [55]. While many believe these conversations are essential in recognizing the autonomy and dignity of the person with ADRD [56], others question whether a person with ADRD will continue to stand by end-of-life decisions they made in the early stages of ADRD once they reach later stages [55]. It was clear that caregivers in the current study were struggling to balance these perspectives, particularly where the care recipients had made it clear they did not want to be a burden on their families. If the person with ADRD feels like a burden to others, this may negatively influence end-of-life decisions and may contribute to thoughts of suicide, homicide, and euthanasia when more dignified alternatives are unavailable [57,58]. 

The present study does have several limitations. While the digital divide is not as large as it once was [59], there are caregivers who may not have regular access to the Internet. This limits the generalizability of the present findings to caregivers who blog. These caregivers may have unique characteristics in relation to their peers who do not blog, including (but not limited to) a larger technological skillset. Given the nature of the data, it is also not possible to confirm whether the bloggers are indeed a caregiver for a person with ADRD. Additionally, demographic information of the sample is limited to what was shared on the blogs by the authors. This makes it impossible to completely ascertain the diversity of the sample or consider additional demographic factors that might have an impact on caregiving and thoughts of suicide and homicide (e.g., the amount of time providing care or specific ADRD diagnosis). This also applies to pre-existing diagnoses of depression and other mood and personality disorders among the caregivers, which could have an impact on the presence of suicidal and homicidal thoughts. Finally, because the initial search for blogs only used the term “caregiver” (a U.S. term), blogs in which people referred to themselves as a “carer” (the term used outside the United States) may potentially have been missed. Because all of the blogs included in the study sample were written by individuals living in Western countries, this may have influenced the themes derived from the analysis and highlights the need for more diverse research in the future, both in terms of suicidal and homicidal ideation and the use of technology by caregivers globally.

Despite these limitations, the present study has numerous strengths. Because caregivers are writing in the moment, there is a lack of recall bias as with traditional data collection methods such as interviews, focus groups, or surveys. The perceived anonymity of the Internet also allows caregivers to express opinions, thoughts, and feelings that they might be unable or unwilling to express in clinical settings or traditional research studies. This makes these and other social media data a valuable and rich resource through which to explore controversial or sensitive topics. 

## 5. Conclusions

Our analysis of blogs written by family caregivers of people with ADRD reveals that thoughts of suicide and homicide may be more common than previous research suggests. These findings underscore not only the complex impact of caregiving on families, but also the need to explore further concepts of suicide and homicide within the caregiving context. In addition to further research, health care providers must consider it a priority to screen for thoughts of suicide and homicide (not just depression) among people with ADRD and their family caregivers, as well as to foster open dialogue regarding end-of-life decisions and advance care planning.

## Figures and Tables

**Table 1 behavsci-09-00022-t001:** Schematic of themes with associated categories and codes.

Theme	Codes and Categories
End-of-life care	loss (*n* = 114, 9/9); death (*n*=73, 9/9); losing PwD (*n* = 67, 9/9); the long goodbye (*n* = 62, 7/9); hospice (*n* = 57, 7/9); only get worse (*n* = 57, 8/9); end of life (*n* = 17, 7/9); funeral (*n* = 2, 1/9); slipping away (*n* = 2, 2/9); good death (*n* = 2, 2/9); worse than death (*n* = 1, 1/9)
Thoughts of death and euthanasia by the person with ADRD	death (*n* = 73, 9/9); want to leave (*n* = 38, 8/9); death of PwD (*n* = 36, 4/9); PwD giving up (*n* = 18, 5/9); ethics (*n* = 17, 3/9); PwD suicide (*n* = 8, 4/9); wanting death (*n* = 7, 3/9); life has no purpose (*n* = 4, 3/9); goodbye to self (*n* = 1, 1/9)
Surrogate decision making	hospice (*n* = 57, 7/9); end-of-life decision making (*n* = 41, 7/9); important documents (*n* = 27, 5/9); end of life (*n* = 17, 7/9)
Thoughts of suicide by the caregiver	caregiver suicide (*n* = 15, 7/9); caregiver giving up (*n* = 10, 5/9); death of caregiver (*n* = 10, 3/9)
Thoughts of homicide and euthanasia by the caregiver	caregiver wanting death of PwD (*n* = 8, 4/9); reference to euthanasia by caregiver (*n* = 4, 2/9)

Note: In addition to the frequency (*n*) of the individual codes used across the dataset, the number of blogs referenced for each code out the full sample is provided in parentheses (e.g., 2/9 indicates two blogs out of the sample of 9). PwD: person with dementia; ADRD: Alzheimer’s disease and related dementia.

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
