# Peer review of "“Like Death is Near”: Expressions of Suicidal and Homicidal Ideation in the Blog Posts of Family Caregivers of People with Dementia"

_behavsci, 2019, doi:10.3390/bs9030022_

Reviewer 1 Report

Thank you for an opportunity to review this well-write, innovative and important paper. It tackles a significant problem and could lead to preventive initiatives to support carers and people living with dementia (PLwD) who may be at risk of suicide/homicide.

A few points which I believe should be addressed:

Could the author present any data on homicide in carers of PLwD? Could they also comment or provide data on murder-suicide incidents or suicide pacts in the study population?

Could the authors provide any data or ideas how the risk of suicide/homicide could be reduced in this population? E.g., support services for carers, respite care, home services for people living with dementia, management of BPSD?

Could they provide some more information about ethical and practical challenges of "conducting primary research on suicidal and homicidal thoughts and behaviours" in the context of the study population (p. 2)?

I am not sure how to read information provided in Table 1 in regards to the frequency of the five major themes. In particular, "all caregivers expressed thoughts related to suicide" (p. 7) - how can I find this number (n=9) in the table?

I also have a question regarding the ethics of conducting online studies like the one described here. Was there a "suicide/homicide risk protocol" approved by the study ethics committee? Was there any expectation of preventing a suicide/homicide if the researchers found indications of an impending suicide/homicide (e.g., detailed plans or a suicide note)?

I am looking forward to reading the revised version of the paper and authors' responses to my comments.   

Author Response

We appreciate the reviewer's comments. Please see our point-by-point response in the attached.

Author Response

(The authors gave the same response as above.)

Reviewer 3 Report

A few comments to improve this qualitative study: There is a need for a graph, schema, or a table that shows the contents . If this is a content analysis, it should be clearly mentioned, with references. A description of content analysis methods and its validity should be given to improve the methods section. This is only an analysis with n of 9. There are more blog posts and bloggers and blogs. That means, the data are heavily nested, and the context impacts their posts. In addition, it is a few individuals who are making the comments. How authors dealt with this issue? Authors need to take strategies by which a few people's voices does not shape the whole paper. The role of gender, age, and education (SES) of the participants / bloggers not known. The charactertics of the relation between the patient and caregiver also needs more information. Age difference? Their genders? Are all participants / couples heterosexual pairs (those who are spouse)?   Implication of the findings are not mentioned. How this study can inform policy and practice and poliy making? How this informs ethical aspects of the cregiving of AD?  These should be discussed in more details. There is also a literature on death anxiety and fear of death that can shape some of the discussions here. Overall, the paper has the potential to contribute, but needs additional work.

Author Response

We appreciate the reviewer's comments. Please see our point-by-point response in the attached.

Round  2

Reviewer 1 Report

Authors have addressed my comments and suggestions. 

Author Response

We appreciate the thorough review of our revised manuscript.

Reviewer 3 Report

OK, now the authors have done their homework and explain to me the exact changes, without asking me to go and read the comments of the other reviewers and the authors' responses to those comments. Ok,good.  That takes care of one of my concerns.  I appreciate it. The authors, however, have declined to bring the death anxiety / fear of death literature. (Please see my last comment and their response to it). They think that the death anxiety literature is very different from their constructs? OK, they should convince us in their paper, not in their response letter. When comes to a reviewer mind may also come to a reader mind. The same is true for a clinician or a researcher who wants to apply these findings to their patient care or to their end of life research. I advise the authors that they should bring such literature either in the introduction or discussion and tell us why they think “Like death is near” is very different from death anxiety. It is to them to make the case for the differences in the meanings and lack of relevance of these topics by providing conceptual, theoretical, or empirical arguments / data. So, we will read such argument and probably agree with them that these constructs are different / irrelevant. But there is a lack of clarity on this issue at the moment, and clearly a difference in the way I see these issues compared to the view of the authors. I think that many of these end of life constructs are related to each other, and views regarding death and its meaning shapes whether they think to homicide or suicide at this life stage.

Author Response

We have added the following to the Discussion, along with the appropriate citations: “Caregivers may experience feelings of distress, depression, anxiety, guilt, and anticipatory grief while providing care for someone with ADRD, particularly when faced with a sense of being unable to cope with the demands of caregiving or during end-of-life decision making. Limited research to date has explored the experiences of caregivers of people with ADRD with regard to death, providing limited insight into aspects of the impact of disease progression, end-of-life decision making and care, and the impact of social support.”

Round  3

Reviewer 3 Report

responses are good and satisfactory.